# A Review on Cyclodextrins/Estrogens Inclusion Complexes

**DOI:** 10.3390/ijms24108780

**Published:** 2023-05-15

**Authors:** Szymon Kamil Araj, Łukasz Szeleszczuk

**Affiliations:** Department of Organic and Physical Chemistry, Faculty of Pharmacy, Medical University of Warsaw, Banacha 1 Str., 02-093 Warsaw, Poland

**Keywords:** cyclodextrin, inclusion complex, estrogen, CD, host-guest complexes

## Abstract

This review focuses on the methods of preparation and biological, physiochemical, and theoretical analysis of the inclusion complexes formed between estrogens and cyclodextrins (CDs). Because estrogens have a low polarity, they can interact with some cyclodextrins’ hydrophobic cavities to create inclusion complexes, if their geometric properties are compatible. For the last forty years, estrogen-CD complexes have been widely applied in several fields for various objectives. For example, CDs have been used as estrogen solubilizers and absorption boosters in pharmaceutical formulations, as well as in chromatographic and electrophoretic procedures for their separation and quantification. Other applications include the removal of the endocrine disruptors from environmental materials, the preparation of the samples for mass spectrometric analysis, or solid-phase extractions based on complex formation with CDs. The aim of this review is to gather the most important outcomes from the works related to this topic, presenting the results of synthesis, in silico, in vitro, and in vivo analysis.

## 1. Introduction

The group of molecules known as steroids are characterized by the presence of a gonane (cyclopentaperhydrophenanthrene) unit in their structures (Figure 1). Many substituents may be added to the gonane unit, which consists of four fused rings made up of three cyclohexanes (A, B, and C rings) and one cyclopentane (D ring). This results in hundreds of different molecules that are present in fungi, animals, and plants. Figure 1 illustrates the first steroid to be complexed by cyclodextrins: ergosterol. Seven Nobel Prizes were given for works on the structure, isolation, synthesis, and biological effects of steroids from 1927 by H.O. Wieland [1] through to 1975 by V. Prelog [2], demonstrating the diversity of steroids and the complexity of their biological roles.

Steroids serve a variety of biological purposes, including, among others: constructing the cell membrane (cholesterol and phytosterols) [3], breaking down fats [4], controlling spermatogenesis [5,6] and secondary gender features (male hormones), increasing muscle mass [7], controlling various reproductive processes [8], contraception [9], and bone density (estrone, estradiol, and estriol) [10,11]. The last three compounds belong to the group of endogenous estrogens; however, the list of compounds with estrogen-like properties is much wider, as will be presented in this review.

In the pharmaceutical industry, cyclodextrins (CDs) are frequently used to improve the aqueous solubility, stability, and bioavailability of medications [12,13,14,15]. Water-insoluble medicines, such as estrogens, may insert part or all of their structure into the cavity of the CD because of its hydrophobic interior, which can lead to the development of host-guest association complexes. In these compounds, there are no covalent interactions between the host CD and the guest molecule, unlike many other complexes. Hydrophobic desolvation, hydrogen bonding, and van der Waals interactions are some of the most significant driving forces for CD complexation [16,17,18], even though the dimensional fit between the host cavity and the guest molecule is crucial for the complex’s formation [19].

The literature regarding the use of cyclodextrins associated with estrogens is vast and diverse (Figure 2); however, to the best of our knowledge, there are no reviews focusing on cyclodextrin-estrogens complexes. Therefore, this article aims to summarize this topic and draw some conclusions resulting from the already published works.

Since the complexes of CDs and estrogens are mainly utilized due to their pharmacological properties, in this review, the list of guest compounds was limited to those grouped in the Anatomical Therapeutic Chemical Classification System (ATC) group G03C [20]. This category includes multiple compounds divided into subgroups such as G03CA Natural and semisynthetic estrogens, G03CB Synthetic estrogens, G03CC Estrogens, and combinations with other drugs and G03CX Other estrogens. Instead of dividing the review of the articles from this topic into parts describing various properties of the complexes, we decided to create separate paragraphs focusing on the particular estrogens. We thought that, in that way, it would be more convenient for the readers interested in a specific estrogen to find all related information on its CD-complexes described in one place. Consequently, each paragraph of the main part of this review is constructed in a similar way, presenting detailed information on the methods of preparation and physiochemical analysis of CD-based complexes of a particular estrogen molecule.

It is well known that the method of complex preparation may have a significant influence on the form of the complex received [21,22,23]. This is why the CD complexes are usually deeply characterized by various analytical methods, both in solid state and solution [24,25,26]. Therefore, the methods and results of the analysis are described in detail in this review. Furthermore, at the end of this work, a synthetic table (Table 1) can be found, presenting direct links to the articles showing the results of the physicochemical analyses of the described complexes.

This paper is divided into three main sections. In the Section 1, the chemistry and role of estrogens and cyclodextrins are briefly summarized. It was not our intention to make this part extensively long, as there have been many recent general reviews focusing on either estrogens [27,28,29,30] or cyclodextrins [31,32,33,34]. The Section 2 is the main one and is divided into paragraphs describing various complexes of particular estrogens. Finally, the Section 7 concludes the paper, but also includes some future perspectives in this topic.

## 2. Estrogens

### 2.1. Chemical Structures of Different Estrogens

As previously outlined in the introduction, the core element of the chemical structure of most estrogens is the gonane (cyclopentaperhydrophenanthrene) unit (Figure 3). However, as has already been proven, this polycyclic system is not necessarily needed for the compound to exhibit the estrogenic activity. Examples of estrogens that do not possess a gonane unit are dienestrol, diethylstilbestrol fosfestrol, hexestrol, zeranol, chlorotrianisene, and methallenestril (Figure 4). The steroid estrogens differ in the number of carbonyl and hydroxyl groups as well as in terms of their location. It should also be noted that some of the steroid estrogens exhibit stereoisomerism and the enantiomers usually significantly differ in their biological activities, such as in the case of 17-α and 17-β estradiol.

### 2.2. Biological Functions and Applications of Estrogens as Therapeutic Agents

Estrogens are a class of steroid hormone that have a wide range of effects on human physiology, including the control of male and female reproductive processes, the development of various tissues, bone integrity, the cardiovascular system, the immune system, and the brain system. These hormones are also linked to the onset or advancement of several cancers [35], osteoporosis [36], neurological [37] and cardiovascular disorders [38], insulin resistance [39], lupus erythematosus [40], endometriosis [41], and obesity [42], as unfavorable consequences.

The three primary endogenous estrogens are 17-β-estradiol (E2), estriol (E3), and estrone (E1), with E2 being the strongest and most prevalent in humans. Another type of estrogen called estetrol (E4) is only produced during pregnancy [43]. The enzyme aromatase converts androgens, particularly testosterone and androstenedione, into all of the abovementioned estrogens, and the liver is where all of these changes mostly take place [44].

The main route of action of estrogens is through the estrogen receptor, a dimeric nuclear protein that possess affinity to DNA, and therefore modulates the expression of genes [45]. Similar to the other steroids, estrogens passively enter into the cell due to their lipophilic nature and bind to the estrogen receptor, causing its activation. As the estrogens can enter all cells, their actions depend on the presence and concentration of the estrogen receptor in the specific cell [46].

Medical applications of estrogens include, among others, hormonal contraception, hormone replacement therapy, and the treatment of gender dysphoria in transgender women as a part of feminizing hormone therapy [47]. The synthetic estrogen derivative of E2, known as 17-α-ethinylestradiol (EE2), is utilized in practically all current formulations of combination oral contraceptive tablets. Other conditions such as menopausal symptoms, breast cancer, and prostate cancer are also treated with it. Many research findings have supported the idea that EE2 might have certain negative consequences, including headaches, breast tenderness, nausea, dizziness, and weight gain. Therefore, EE2 is often advised for oral administration at a lower doses to avoid its side effects [48,49].

EE2, like most estrogens, belongs to the class II of the Biopharmaceutical Classification System (BCS). Active pharmaceutical ingredients (APIs) from this group are characterized by high permeability, but also low solubility. Therefore, the bioavailability of those products is limited by their dissolution rate. According to previous studies, the bioavailability of EE2 is directly proportional to the water solubility of the drug [50,51]. Therefore, various methods such as cocrystals formation or the application of nanocarriers are utilized to boost the EE2 dissolution rate [52,53]. Moreover, the clearance of EE2 measured after intravenous administration is weaker than that measured after oral administration of the same dose of EE2, and the absorption value of EE2 after injectable administration is higher than that of oral administration [54]. Complexes of various CDs with EE2 are used to obtain drugs in various dosage forms.

### 2.3. Endocrine Disrupting Chemicals (EDCs)

Both endogenous estrogens, as well as the artificial ones from contraceptives and animal growth agents, enter into the environment through the metabolism and excretion of humans and animals. Once in the environment, estrogens contribute, amongst other compounds such as bisphenol, octyl phenol, or nonylphenol, to a group of endocrine disrupting chemicals (EDCs). These compounds are typical pollutants in water systems, exhibiting strong estrogenic potencies and eco-toxicity [28,55,56,57].

Even at very low concentrations, EDCs can cause serious endocrine disorders. For example, in water environments, EDCs have been attributed to reproductive disturbances in wildlife, such as the feminization of male fish [58,59]. Due to their physicochemical properties, such as lipophilicity and neutral character, estrogen pollutants are difficult to remove and they gradually accumulate in water systems, posing a great risk to ecological security [60]. Furthermore, freshwater worms can bioaccumulate EDCs, making a transfer to benthivores possible and the subsequent secondary poisoning of predators. In addition, synergistic effects can be observed between some EDCs, inducing greater harm and increasing the risk of poisoning [61]. In this regard, the purification and sanitation of estrogen pollutants is a critical issue worldwide. As will be shown in the following sections, CDs can be of particular help regarding this topic.

## 3. Cyclodextrins and Their Inclusion Complexes with Steroids

A hydrophobic cavity that can enclose other substances makes up the heart of the structure of cyclodextrins (CDs), a class of cage molecules formed from α-D-glucopyranose units. Because of its exceptional encapsulation qualities, a “host-guest” connection develops, which can change or enhance the guest molecule’s physical, chemical, or biological features. While natural α, β, and γ cyclodextrins, composed of six, seven, and eight glucose subunits, respectively, are the most widely used ones, large-ring cyclodextrins (LR-CD), which have nine to more than several hundred units, have also been researched and employed [62,63]. In addition, due to their special qualities, derivatives of native (non-substituted) CDs have found significant use in a variety of industries, including pharmaceuticals, cosmetics, biomedicines, textiles, and the food industry [64,65,66].

Polymer additives and CDs covalently bonded to polymers are now of interest in addition to two-component inclusion complexes made up of a guest molecule and a CD. For example, molecularly imprinted polymers are based on CDs resulting from the non-covalent interactions between the guest molecule (template) and the monomer in the presence of a cross-linking agent during polymerization. The template gives nanosponges the ability to recognize particular molecules. In those and other complex systems, not only the interactions occurring inside the CD’s cavity, but also the intermolecular and intramolecular forces including the outer part of the CD, play an important role in the stabilization of such materials. Every year, new uses for CDs and CD-based materials are found because to the biodegradability, biocompatibility, and adaptability of these media [67,68,69]. Pharmaceutical formulations frequently include CDs because they improve the dissolution of APIs with low solubility and protect molecules from environmental influences, including light, humidity, and heat [12,13,70]. The disagreeable taste or odors of drugs can be concealed by CDs, which is crucial for formulations intended for young patients [71]. The first pharmaceutical product containing CD, prostaglandin E2/β-CD sublingual tablets (Prostarmon E, Ono), was marketed in Japan in 1976 [72]. Currently, CDs are used as excipients in the manufacturing of more than 100 unique pharmaceuticals [73,74,75].

Steroids are ideal guest molecules for complexation by cyclodextrins, due to their hydrophobic nature and size, which is comparable with one of CD’s cavities. Therefore, a lot of hydrophobic interactions, stabilizing these kind of complexes, can occur between most of the steroids and the majority of CDs. The very first publication on cyclodextrin/steroid complexes was most probably that by Cramer and Henglein (1957) [76], showing the increase in solubility of ergosterol (Figure 1) when in a complexed form with beta-cyclodextrin at a 1:2 molar ratio. Cyclodextrin-steroid interactions have received a great deal of attention in the 60 years after Cramer’s initial research. The most extensively researched steroid in the literature on cyclodextrin is cholesterol, followed by steroid medications, including estrogens, and bile acids with their salts. Several publications have shown the enhanced selectivity of HPLC methods for steroids when CDs are employed as a chiral stationary phase or in the mobile phase [77,78]. Recent studies on capillary electrophoresis have shown the advantage of CDs in the run buffer to aid separation of closely eluting compounds [79,80].

## 4. Estrogens That form Host-Guest Complexes with Cyclodextrins

### 4.1. Ethinylestradiol (EE2)

#### 4.1.1. Complex Preparation Methods

The preparation of EE2 (Figure 3) and cyclodextrin complexes usually occurs in a methanol/water environment. It is important to note that the ratio between MeOH and water causes marked association constant values differences. Studies have shown that association constants in water are higher than in methanol [81]. Because of the high logK_ow_ = 4.12 of EE2, the usage of surfactants (sodium dodecyl sulfate, SDS) and buffering agents to maintain pH = 11.5 is helpful in this case [82]. The reactants for the complexation reaction usually include water/methanol solutions of EE2 and CDs. The solution composed of methanol, water, and EE2 should be stored at 6.5 °C, with the exclusion of the evaporation of methanol and without the presence of light before the reaction. The procedure for preparing the CD solution for the reaction is quite similar; CD should also be dissolved in water with methanol (usually at a 45:55 ratio). There are known cases of using ethanol instead of methanol; i.e., 1 mL of pure EtOH to 10 mL of deionized water. The reaction takes place by mixing the two reagents and stirring them [83]. There was no information found regarding the preparation of high concentration complexes. The stoichiometry of the reaction shows that the formation of complexes can occur, depending on the CD used, at a ratio of 1:1 [81,84,85], 2:1 [86], 1:2 (only for γ-CD, because it has the largest cavity) [87], or a mix of these values.

Studies have shown that the usage of β-CD to form complexes with EE2 can cause difficulties because its cavity may be too small to fully bind with the hydrophobic elements of this steroid. That is why the usage of γ-CD or 2HP-γ-CD is more effective, due to their larger sizes. The association constant of EE2 with γ-CD in methanol-water (45:55, *v*/*v*) at 35 °C is about twice as high as with β-CD [88] (or even more, as shown in other studies [82]). The association constants in an acetonitrile-water environment are lower than in methanol-water solutions [89]. Even while working with γ-CD, one can encounter obstacles. According to the comparison of EE2 with γ-CD and 2HP-γ-CD complexes by Shakalisava and Regan [82], despite the application of CDs with larger cavities, the steroid can still have difficulties in forming hydrogen bonds between substituted hydroxyl groups and itself. The same can occur in cases of other estrogens, such as equilin, E3, E2, and E1 [82].

#### 4.1.2. Complex Structure Analysis

To study their structure and properties, the obtained complexes have been analyzed using many methods. While association constants have been usually determined using HPLC, one paper also showed association constants of EE2/β-CD and EE2/γ-CD fluorometrically measured in methanol/water (20/80 *v*/*v*) at 35 °C [90]. Then, the values were compared to corresponding ones obtained from an another HPLC study. The values from fluorometric determination were found to be lower, which was, however, not fully explained by the authors [91]. The solubility of pure EE2 in water is very low, approximately 11.3 mg L^−1^. Gong et al. showed that complexation with diethylenetriamine-β-cyclodextrin (DETA-β-CD) increases the solubility more than forty times to approximately 496 mg L^−1^, which was the highest value among the analyzed types of CD in this study [83]. The surface morphology and composition of complexes have been analyzed using scanning electron microscopy (SEM). It confirmed the difference between the EE2 and CD physical mixture and the complex, showing a homogeneous and compact plate-like structure crystal [83]. To prove that van der Waals interactions occur in the cavity of CD, UV spectroscopy has been applied [83].

To estimate the alteration in the microenvironment around protons, resulting from the formation of the inclusion complexes, ^1^H NMR spectroscopy has been successfully used [83]. ROESY and NOESY are NMR techniques, based on the Nuclear Overhauser Effect, through which space interactions are observed between protons of the host and the guest molecules if their internuclear distance is smaller than 4 Å. Those techniques have also found their application in the analysis of EE2/CD complexes [83]. Based on the NMR results, it has been found that the EE2 A ring penetrates the β-CD cavity narrow rim and the D ring is located close to the β-CD wide rim [83]. This was, however, only valid for the complexes formed in the 1:1 host:guest ratio. As a result of the NMR analysis, it was also found that, when using β-CD, the C≡CH group of the D-ring locates inside the cavity, without protruding the primary rim, while in the case of complexation with γ-CD, the rings of the steroid can penetrate the cavity further [92].

The CD/EE2 complexes have been also studied using the in silico methods. For example, in one of the studies, molecular docking revealed that in the 2:1 host:guest complexes, the cavity of the second CD is penetrated by the D ring of EE2, while the A ring is more favorable to be included in the β-CD cavity. The structure and dynamics of HP-β-CD/EE2 and β-CD/EE2 complexes were studied using MD-simulations [85]. 

Valuable results have been obtained by studying CD/EE2 complexes using fluorescence spectroscopy. For example, it has been shown that the presence of β-CD or DM-β-CD increases the fluorescence intensity of EE2. This allowed for an evaluation of the stoichiometry and stability of the complexes [90]. In another study employing fluorescence spectra measurements, it was found that the EE2 molecule complexed in the β-CD cavity has a significantly reduced rotational freedom and vibrational level relaxation (VR) [81]. 

The thermodynamic properties of the complexes formed by the EE2 and CDs have been also determined in a few cases. In one study, the authors studied how the solvent affects the enthalpy and entropy of the formation of β-CD/EE2 and γ-CD/EE2. The results showed that the changes in enthalpy and entropy were more negative for each complex in a methanol-water environment than in acetonitrile-water mixtures [89]. This was explained by the structural differences of those complexes formed in various solvents, indicating a significant role of the solvent in the intermolecular interactions occurring between the guest and host molecules. There have also been known cases of measuring the influence of CD on the pK_a_ values of EE2 [90]. The change in the acidity constant of EE2, caused by the presence of cyclodextrins, allowed the authors to infer the interaction between the phenolic hydroxyl of the A ring of the EE2 and the external groups of the CDs. These experimental results have been supported by molecular modeling calculation results, indicating that the dipole moment and the conformation of EE2 changes due to complexation with β-CD and HP-β-CD. As anticipated, the conformation of CDs, especially HP-β-CD, changes as well upon complexation [85].

#### 4.1.3. Application of the Complexes

Cyclodextrins, by forming inclusion complexes, enhance the water solubility of EE2. For example, the β-CD/EE2 complex has found its application as an API. There are known cases of application of β-CD/EE2 complexes in commercial drugs such as Yaz^®^ [93], Safyral^®^, Beyaz^®^, and Lorina^®^ manufactured by Bayer^®^ or Sandoz^®^ [86]. While in these aforementioned cases the route of administration is oral, formulations of transmucosal and buccal routes of administrations of the hormone-complex, more precisely as a PVA-film, have been developed as well [86]. This form was chosen to ensure the homogenous formulation of the API. In situ gelling nasal inserts were developed by adding methyl β-CD to carrageenan (sulfated polysaccharides from seaweeds) to ensure the slower absorption of EE2, with a lower peak serum level compared to the nasal spray (Aerodiol^®^). In another study, conducted to compare the dissolution rates of “raw” EE2 and DETA-β-CD/EE2 in simulated blood, clearly showed the advantage of the complexed form of this API [83]. However, in a similar study, it was found that the complexation does not affect the pharmacokinetics and relative bioavailability of ethinyl estradiol. This was further confirmed by a study on 18 healthy postmenopausal women receiving EE2 noncomplexed or complexed with β-CD as a single oral dose [94].

In addition to their applications as a drugs, CD/EE2 complexes have also been applied in water purifying technology. According to Fenyvesi’s study, in a small scale laboratory water purifying experiment using CD polymer filters, the amount of EE2 decreased by 87–99% [86]. The use of CD can be also a suitable strategy to measure the concentration of EE2 at such low levels as ppb, without using either derivatization reactions or toxic organic solvents [90]. A more specific study was performed using β-CD and γ-CD in water at pH = 7. It showed that γ-CD is more efficient in removing EE2 pollution, due to its faster ability to achieve adsorption saturation and higher adsorption capacity. In this experiment, the amount of EE2 decreased by 90% within 30 s. It is also known that re-using the CD-filter is possible. To regenerate the CD-polymer filter, it has to be soaked in ethanol or another alcohol that dissolves EE2 well. This leads to a 99% efficient filter recovery [87]. The usage of CD in micropollutant removal seemed to be very promising, so larger scale experiments were performed. β-CD crosslinked with epichlorohydrin was used for the preparation of the filter. The results showed a correlation between the binding constant of the complex and the sorption efficacy within the filter. The concentration of EE2 (5 μg/L) was lowered by nearly 100% after 5 min. Longer observations (2 h and 48 h) showed no further changes in concentration. In the same article, it was shown that the CD filters were highly selective in removing selected micropollutants. This makes CD a suitable technology for wastewater management [95]. β-CD crosslinked with epichlorohydrin and mixed with quartz sand, active carbon, or commercial sorbent was also studied on a smaller, lab scale. The results confirmed a high utility of CD in water waste treatment [96].

### 4.2. Estradiol (E2)

#### 4.2.1. Complex Preparation Methods

The process of the formulation of E2/CD complexes is similar to EE2/CD. It can be achieved by adding an excess amount of E2 (Figure 3) to a CD solution. To allow the complexes to be formed, the solution should be stirred for 24 h [97]. Storing the complex for further use should be carried out at low temperature; for example, at 4 °C [98]. The current state of knowledge regarding the stoichiometry of the complexes indicates a 1:1 [98,99,100] value for HP-β-CD and β-CD or a 2:1 [99,100] for β-CD in the form of polymer-based materials. The 1:1 stoichiometry with β-CD was also shown to be stable in a molecular modeling study [101]. Both native and modified β-CDs are more effective in forming complexes with E2 in aqueous solutions than α-CD or γ-CD [102].

#### 4.2.2. Complex Structure Analysis

The structure of the complexes of E2 with various CDs has been extensively studied. It has been generally approved that the complex is formed by the penetrating of the cyclodextrin’s cavity by the estradiol molecule. In a recent work [100], the authors claimed to successfully solve the crystal structure of the β-CD/E2 complex. However, after a closer look, we found that, in this study, it was only possible to determine the positions of the CD’s atoms, while the orientations of E2 molecules were not established due to the high level of structural disorder. In the same article, the structure and stability of the β-CD/E2 complex have been studied using many different methods, such as DSC, NMR, thermogravimetric analysis, and hot stage microscopy [100].

In another theoretical work, the representative geometries of β-CD/E2 and HP-βCD/E2 were obtained using molecular dynamics simulations. The in silico study showed that the aromatic ring was located closer to the wider rim and the cyclopentyl to the narrow rim of the CD. The association constants of β-CD/E2 and HP-β-CD/E2 were established at 298 K [101]. Another paper also showed the association constants of E2/β-CD and E2/γ-CD fluorometrically measured in methanol/water (20/80 *v*/*v*) at 35 °C. Then, the values were compared to values from an HPLC study. The values from fluorometric determination were found to be lower than those obtained by the chromatographic method [91]. In another HPLC study, the values of retention factors were determined for the complexes formed between the E2 and α-, β-, and γ-CDs, in their native and hydroxypropyl derivatives forms at temperatures from 0 to 60 °C. The retention factor of the complex decreases with the increase in the number of the sugar molecules in the cyclodextrin ring. The same study pointed out that the high retention factor reflects the strength of the host-guest interactions and highlights the importance of temperature in chromatographic complex separations [103]. The presence of β-CD makes it easier to separate E2 from other estrogens using HPLC at specific temperatures. This is caused by shorter retention times for this estrogen [104]. When E1, E2 (both stereoisomers), E3, and E4 have to be separated using HPLC, the addition of β-CD and a temperature at exactly 26 or 47 °C enables excellent separation [105]. Another chromatographic method—micellar electrokinetic chromatography (MEKC)—also showed an enhancing effect on estrogen separations using β-CD and γ-CD as additives. Thanks to the addition of CDs, E2 was more easily separated from another nine different estrogens. A separation using α-CD was also performed, but the results showed no successful separation in that case. This can be explained by the cavity of α-CD not being large enough to successfully bind with the estrogens [106]. DSC experiments showed that the characteristic endothermic curve peaks of E2 at 179.60 °C disappear from the curve after complexation with β-CD [99]. The thermodynamic parameters of the complex formation were also measured and analyzed. This determined that the dominating interaction forces are the hydrophobic and hydrogen-bonding interactions, which stabilize the β-CD/E2 complex, as well as the hydrogen bonding interaction and van der Waals forces for the HP-β-CD/E2 complex [101].

SBE-β-CD/E2 has been widely studied in silico using the semi-empirical PM3 quantum chemistry method for 17α- and 17β-estradiol isomers. The SBE-β-CD is a β-CD derivative with either two or four sulfobuthyl ether groups substituted. In the case of four groups, their orientation can be different and have a significant impact on the complexation. These groups can be connected to the CD in two orientations—“up” or “sideways”. The variant with no groups in the “up” orientation forms the most stable complex. Other variants are unfavorable for the complexation. Heat of formation, dipole moments, stabilization energies, and the capillary electrophoresis migration time for PM3-optimized geometry of the E2 inclusion complexes were calculated. A correlation has been found between the migration time and the heat of formation. It suggests that more stable complexes have a longer migration time in capillary electrophoresis. Furthermore, the theoretical in silico calculations results indicate that a CD/E2 complex may exist with the A or D ring in the cavity while migrating [107].

Another study showed interesting properties of HP-β-CD/E2 complexes. Liposomes filled with the complex can successfully prevent the induction of MISS (membrane-initiated steroid signaling) effects, occurring when the non-complexed E2 is being applied. Furthermore, liposomes filled with the complexes were as efficient as free E2 to promote uterotrophic effects in mice in in vivo experiments. Results describing the size and morphology of the liposome-CD-estrogen complexes were obtained using TEM [97]. Estradiol is a sex-steroid scarcely connected with female gender. It is interesting that while coexisting with testosterone in an HTG-β-CD and HTMT-β-CD solution, E2 exclusively forms an inclusion complex with HTG-cyclodextrin, while testosterone couples with HTMT-β-CD. This might be due to the absence or presence of the methyl group 19 at carbon 10 for E2 or testosterone [108].

#### 4.2.3. Increased Solubility Resulting from Complexation

The complexation of E2 with various cyclodextrins leads to the higher water solubility of this steroid. In a study comparing the dissolution enhancement caused by different CDs, HSES-β-CD had the best influence on this parameter among the six studied CD derivatives. Another work determined HP-β-CD to have the best solubility increasing properties among the cyclodextrins studied [102]. A higher solubility of E2, caused by the complexation with CDs, can lead to reduced doses of this API in the drug formulations in steroidal E2 therapies, increasing patient safety and decreasing the cost of the therapy [108]. Apart from the oral route of administration, the complexation also has an enhancing effect on the through-skin E2 permeation. This was observed in an in vitro study, using Franz diffusion cells. A rising flux rate was also noted. This forms the conclusion that the complexation of E2 with CDs increases its bioavailability through the skin. In the same study, the solubility of E2 in the randomly methylated beta cyclodextrin complex (RM-β-CD/E2) was also measured. It showed an increase in solubility compared to HP-β-CD complexes [109]. Another in vitro dissolution test compared the “simple” β-CD/E2 complexes with the form of polymer-based beads and the irregularly-structured polymolecules of β-CD/E2. The medium in this test was FBS (fetal bovine serum). Complexes in the form of beads have a five times higher dissolution rate (15%) after 2 h than irregularly shaped ones. In an in vitro permeability study, using Caco-2 cells, a β-CD/E2 had much lower permeability values than pure E2, used in this experiment as the control [99].

#### 4.2.4. In Vitro and In Vivo Analysis

One study showed the effects of E2 on *Danio rerio* depending on the formulation of the steroid. The research confirmed that CD/E2 has a less teratogenic influence on fish embryos than free E2. The adult individuals, while being exposed to E2 instead of E2/β-CD, but at the same dose of estrogen, presented a decrease in moving activity and a more aggressive behavior. The results of this study showed that E2/β-CD is a less toxic form because E2 is also transported through the cell membrane by diffusion and the hydrophilicity of the complex makes this process unlikely to happen [98]. This makes it more difficult to affect the estrogen receptors, which are located in the cytoplasm of the cell. Another group of scientists tried to overcome this problem by measuring the biological properties of Amino-CD with amine groups connected around the cyclodextrin. They synthesized many amino derivatives using click chemistry. They discovered, using confocal ultra violet microscopy, that a complex labeled with rhodamine stopped on the nucleus membrane after penetrating the outside one. The steroid influenced the cell by activating its cytoplasmic estrogen receptors [110]. There was also another study, demonstrating a comparison of the binding affinities of pure E2 with estrogen receptors and a complexed one, which resulted in value differences that were no bigger [102]. In an in vitro study, where the influence of E2 encapsulated in HP-β-CD on mice zygotes was studied, showed that the complexation leads to higher activity on mice zygotes. Very low concentrations led to an inhibition of blastocyst formation. To verify the influence of an complex on the inhibition, similar experiments have been carried out with pure CD and pure E2 [111]. For instance, β-CD in various forms (irregular and microbeads) showed no toxic effect on Caco-2 cells [99]. E2 is also a precursor for biotransformation into 2HE2 and 4HE2. *Mucuna pruriens*, a tropical legume native to Africa and tropical Asia, cell cultures can perform this biotransformation using phenoloxidase enzymes. To enhance the efficiency of this process, the water solubility of E2 had to be increased. β-CD/E2 were the natural choice. The complexation was confirmed using DSC. The complexation did not stop the bioconversion and resulted in higher amounts of products. The final products of biotransformation can be obtained using appropriate organic solvents to extract them from β-CD [112,113].

#### 4.2.5. Application of the Complexes

CD complexes can be used in the formulation of E2 controlled release drugs. The preparation of a hydrogel by combining CDs and hydroxypropyl methylcellulose (HPMC) can lead to a prolonged E2 release, even lasting several days. This solution also enables an increase in the E2 solubility per gram of API, which leads to dose reduction and lowers the cost of the therapy. The preparation of and tests on the stability of such systems have been performed with various complexes and under different conditions. For example, the solubility of CD/E2 before and after autoclaving was measured. It turned out that β-CD/E2 and SB-β-CD/E2 are a bad choice for this purpose because the cross-linking process is more difficult for both [114]. The preparation of nasal inserts often uses carrageenan as an insert-forming substance. A study showed that the presence of M-β-CD/E2 complexes, instead of pure E2, had an big influence on the viscosity of such solutions. E2 has much better biopharmaceutical properties in in situ nasal inserts when complexed with CD, such as a longer period without drug concentration peaks, reduced bioadhesion of the insert, and increased moisture sorption [115].

E2, in the form of a complex with RM-β-CD (randomly methylated β-cyclodextrin), can be found in a commercial drug, Aerodiol^®^, produced by Servier (Europe) [93,108]. E2/CD can also be used in transdermal formulations, for example, in the form of the β-CD complex mixed with a Pentravan^®^ vehicle, which is available for patients from an compounding pharmacy [98].

E2 can be encountered as a water micropollutant, so the usage of CD in E2 removal has been studied on a large scale. For example, in one of those studies, β-CD crosslinked with epichlorohydrin was used for the preparation of the filter. The results showed a correlation between the binding constant of the complex and the sorption efficacy within the filter. The concentration of E2 (5 μg/L) was lowered by nearly 95% after 5 min of treatment. Longer time observations (2 h and 48 h) showed a reduction in the level of almost 100%. The same paper showed that CD filters were highly selective in removing selected micropollutants. This makes CD a suitable technology for wastewater management [95]. β-CD crosslinked with epichlorohydrin and mixed with quartz sand, active carbon, or commercial sorbent was also studied on an smaller, lab scale. The results showed a high utility of CD in water waste treatment [96]. Another form of cyclodextrins were used to lower the pollution level of E2. In another study, the β-CD-coated micro magnetic activated carbon was used for this purpose, resulting in the successful recovery of E2. In addition, this absorbent can be used up to eight times without losing its properties. This proves that magnetic solid-phase microextraction is also helpful in E2 water pollution handling [77].

### 4.3. Estradiol Derivatives

2ME2 (Figure 3) forms stable complexes with CDs, usually in a 1:1 stoichiometry. A UV spectroscopy method was used to study the stability of the 2ME2 complexes with DM-β-CD and TM-β-CD. In these cases, the inclusion occurs by the penetration of the D ring of 2ME2 to form the secondary side of the CD while the A and part of the B ring are protruding from the secondary side of the CD. These complexes have been extensively studied using many methods, including DSC, HSM, TGA, FTIR-spectroscopy, UV-spectroscopy, X-ray, and molecular modelling. While DM-β-CD/2ME2 forms monoclinic crystals and TM-β-CD/2ME2 triclinic ones, both complexes showed an enhancement of the water solubility of 2ME. The solubility enhancement factors were compared within a group of 10 cyclodextrins. It was shown that TM-β-CD/2ME2 had the highest solubility enhancement factor value, nearly 25 higher than β-CD [116].

SBE-β-CD/2ME2 is a CD complex that has been studied in silico using the PM3 semi-empirical method. The SBE-β-CD was determined as a CD derivative with either two or four sulfobuthyl ether groups. In the case of four groups, their orientation can be different and have an significant impact on the complexation. These groups can be connected to the CD in two orientations—“up” or “sideways”. The variant with no groups in the “up” orientation forms the most stable complex. Other variants are unfavorable for the complexation. The heat of formation, dipole moments, stabilization energies, and the migration time (capillary electrophoresis) for the PM3-optimized geometry of the 2-hydroxyestrone inclusion complexes were calculated. A correlation between migration time and the total heat of formation occurred. It suggests that more stable complexes have a longer migration time in capillary electrophoresis. Furthermore, the theoretical in silica calculations suggest that a β-CD/2ME2 complex may exist with the A or D ring in the cavity while migrating [107].

The MEKC separation of ten different estrogens, including 2HE2, 4HE2, and 16KE2, was performed in the presence of α-, β- and γ-CD. The addition of α-CD did not improve the separation. However, the β-CD and γ-CD made the separation more effective. Furthermore, 16KE2 was one of the two studied estrogens (the other was 16HE1), for which the addition of CD impacted in a reversion of the migration order. The addition of γ-CD showed the best results, and it is highly recommended to add the γ-CD to MEKC separations of various estrogen mixtures [106].

In addition, 4HE2 and 2HE2 can be obtained as products of an E2 bioconversion using phenoloxidase in *mucuna pruriens* cell cultures. An E2/β-CD complex was used to enhance the concentration of E2 in water before the reaction. The bioconversion was successfully performed. The fact that E2 was inside the CD cavity during the reaction did not affect the structure of the products. The products of the reaction, 2HE2 and 4HE2 (with the majority of 4HE2), were still complexed with CD after the reaction was completed. This was confirmed using DSC [112,113].

### 4.4. Estrone (E1)

#### 4.4.1. Structure of the Complexes and Complex Formation Mechanism

Estrone (Figure 3) is a steroidal estrogen and, like most of the other compounds from this group, it forms stable complexes with various CDs, including both native α, β, and γ CDs as well as their functionalized derivatives. These host molecules significantly differ not only in terms of the size of their cavities, but also in their solubilities, lipophilicity, and their abilities to form the inter- and intramolecular forces. The stoichiometry of such complexes is usually determined to be 1:1 [84,88]. The stoichiometry of 1:1 with six various cyclodextrins was confirmed during an estimation of their association constant values using a relationship analogous to the Benesi–Hildebrand equation [90]. Another study confirmed the 1:1 stoichiometry using the Hummel-Dreyer method [88].

E1 can enter the cavity of β-CD by either the A-ring or the D-ring. This is possible because of the weak steric hindrance of the molecule with the CD. Such conclusions were obtained from the molecular docking results. In addition, the calculations allowed for an estimation of the stability constants of the E1 complex with β-CD and γ-CD in two possible orientations, the A-ring or D-ring in the cavity. The calculated K_A−up_ was greater the larger the K_D−up_, which indicates that the A-ring penetrating the cavity is more likely to occur [92]. The association constant of E1/β-CD was measured using HPLC in methanol/H_2_O 45:55 *v*/*v* solution at 35 °C. It was shown that the effect of adding the CD to increase the solubility is much weaker for E1 than it was for E2, EE2, or E3 [84]. What was unique for E1, in comparison with other estrogens, was that γ-CD/E1 was characterized by a higher association constant than β-CD/E1 [88]. This was also confirmed in another study with γ-CD in the native form and using its derivatives [82]. The complexation constant was determined using the spectrofluorimetric method for five other CD complexes with E1 in water and other solvents as well [90].

#### 4.4.2. Methods of Complex Preparation

It has been shown multiple times that the solvent has an major influence on the association constant value for E1/CD complexes. For example, in MeCN/water solutions, the value of the complexation constant is nearly three times lower than in the MeOH/water environment [89]. In another study, the fluorescence spectra of complexes formed between E1 and various CDs were measured in aqueous solution at neutral pH. The results confirmed the formation of inclusion complexes between E1 and all of the studied CD derivatives, except for α-CD and S-β-CD [90]. The enthalpy and entropy of formation values for γ-CD/E1 and β-CD/E1 were determined in two different environments [89]. The retention factors for α-, β-, and γ-CD/E3 complexes and their HP-CD derivatives were measured using HPLC at 0 and 60 °C in MeCN/water solution (35% *v*/*v*). The differences between the retention factors point out the importance of temperature control in the HPLC analysis of those complexes. γ-CD/E3 and its HP-derivative were the most effective host-guest complexation [103].

#### 4.4.3. Application of the Complexes

SBE-β-CD/E1 was extensively studied in silico using the PM3 semi-empirical method. It was shown that increasing the number of sulfobuthyl substituents decreases the stability of the formed complex. Heat of formation, dipole moments, stabilization energies, and the capillary electrophoresis migration time for PM3-optimized geometry of the E1 inclusion complex were calculated. A correlation occurred between migration time and total heat of formation. It suggests that more stable complexes have a longer migration time in capillary electrophoresis. The theoretical calculations also suggest that the CD/E1 complex may exist with either the A or D ring in the cavity while migrating [107].

The presence of β-CD facilitates the separation of E1 from other estrogens using HPLC at specific temperatures. This is caused by the reduction in the retention time caused by adding the CD [104]. When E1, E2 (both stereoisomers), E3, and E4 have to be separated using HPLC, the addition of β-CD and a temperature at exactly 26 or 47 °C enables excellent separation [105]. A similar study confirmed the usefulness of β-CD and two other derivatives in E1 HPLC separations with very similar estrogens, such as equilin, 2HE, 4HE, or 16HE. The influence of the concentration of the CD in the mobile phase was also studied [117].

Similarly, as for E2, an enhancing effect on estrogen separations in MEKC using β-CD and γ-CD as additives was shown for E1. Due to the addition of CD, E1 was more easily separated from another nine different estrogens. A separation using α-CD was also performed and the results showed no successful separation. This could be caused by the cavity of α-CD not being large enough to efficiently guest estrogens [106].

E1-based molecules have found their application in female breast cancer therapy. A novel estrogen-anchored cyclodextrin drug complex (CDE1-Ada-DOX) was prepared by connecting E1 through an nitrogen-carbon bond to the side of a β-CD molecule, while the cavity of CD was occupied by a doxorubicin derivative. The intramolecular self-assembly properties of the modified CD were extensively studied by means of NMR, SEM, and TEM. The obtained results demonstrated the targeted therapeutics delivery of CDE1-Ada-DOX to breast cancer cells in a controlled manner and that the drug vector CDE1 could be potentially employed as a molecular tool to differentiate nongenomic from genomic mechanisms [118].

CD/E1 complexes have also been the subjects of studies focused on the water pollution problem. A recent work showed that β-CD- and γ-CD-based polymers can form highly effective estrogen filters. In this work, the authors synthesized mesoporous β-CDP and γ-CDP by crosslinking CD with rigid aromatic groups, increasing their surface area. The γ-CD-based ones showed better results in terms of the removal of E1 from water [87]. Other forms of cyclodextrin-based materials were used to lower the water pollution level at a laboratory scale. β-CD-coated micro magnetic activated carbon was usefully applied in this case. The recovery of E1 was successful and the absorbent could be used up to eight times. This proved that magnetic solid-phase microextraction is also helpful in E1 water pollution handling [77].

### 4.5. Estrone Derivatives

SBE-β-CD/2HE1 has only been studied by means of molecular modeling methods so far. Quantum-mechanical studies calculations have been used to determine the heat of formation, dipole moments, stabilization energies, and the capillary electrophoresis migration time for PM3-optimized geometry of this complex. A correlation between migration time and the total heat of formation occurred. It suggested that more stable complexes have a longer migration time in capillary electrophoresis. Furthermore, the theoretical calculations suggest that a CD/2HE1 complex may exist with either the A or D ring in the cavity while migrating [107].

The separation of estrogens may be very difficult. To facilitate this task, the influence of β-CD, DM-β-CD, and HE-β-CD on the separation of 2HE1, 4HE1, or 16HE1 using HPLC was studied. The addition of CD was found to make the separation much more effective. The influence of concentration on the efficiency of this process was also studied. β-CD derivatives are especially recommended for successful HPLC separation [117]. The MEKC separation of ten different estrogens, including 4HE1, 16HE1, and 2ME1, was performed in the presence of α-, β-, and γ-CD. The addition of α-CD did not improve the separation. However, the β-CD and γ-CD made the separation more effective. Furthermore, 16HE1 was one of two studied estrogens (the other was 16KE2) where the addition of CD impacted in reversion of the migration order. The addition of γ-CD showed the best results, and it is highly recommended to add the γ-CD to MEKC separations in various estrogens mixtures [106].

### 4.6. Estriol (E3)

#### 4.6.1. Complex Preparation and Structural Studies

Estriol (E3) (Figure 3) is another example of steroidal estrogen. Like other compounds from this group, the most stable complexes of E3 are those with β-CD and its derivatives. The stoichiometry of such complexes is usually determined as 1:1 [84,88]. The stoichiometry of 1:1 with six various cyclodextrins was confirmed during an estimation of their association constant values using a relationship analogous to the Benesi–Hildebrand equation [90]. Another study confirmed the 1:1 stoichiometry by using the Hummel-Dreyer method [88].

Estriol strongly binds with the β-CD by fully penetrating its cavity, which was studied using the solution ^1^H NMR technique [119]. Furthermore, as shown by the molecular docking calculations, E3 can enter the cavity of β-CD either by the A ring or the D ring, resulting in both orientations of host-guest complexes present in the solution. This is possible because of the weak steric hindrance of the molecule in the CD, which is also true for E1. The calculations also estimated the stability constants of the E3 complexes with β-CD and γ-CD in two possible orientations, with either the A ring or the D ring penetrating the cavity. ln K_A−up_ was found to be larger than ln K_D−up_, which indicates that the A-ring penetrating the cavity is more likely to occur [92]. The association constant (K_a_) of E3/β-CD was measured using HPLC in methanol/H_2_O 45:55 *v*/*v* solution at 35 °C. The complex had a much lower value of K_a_ than E2. For E1 and EE2, K_a_ was at a very similar level [84]. The Hummel-Dreyer method was also used to estimate the K_a_ in methanol/water solution. It showed major differences of this constant value among three studied estrogens, with the E3 possessing the highest one. γ-CD/E3 had a higher association constant than β-CD/E3. This was also confirmed in another study also on γ-CD derivatives [82]. The same constant was estimated using a spectrofluorimetric study for five other CD complexes with E3 in water, as well as in many other solvents [90]. Another study also showed the association constants of E3/β-CD and E3/γ-CD fluorometrically measured in methanol/water (20/80 *v*/*v*) at 35 °C. Then, the values were compared to corresponding ones from an HPLC study. The values from fluorometric determination were lower [91]. The solvent has an major influence on the K_a_ value. For example, in MeCN/water solutions, this value is nearly three times smaller than in a MeOH/water environment [89].

The fluorescence spectra of complexes formed with various CDs were recorded in aqueous solution at neutral pH. The results confirmed the formation of inclusion complexes with all studied CD derivatives, except for α-CD and S-β-CD [90]. The enthalpy and entropy of the formation of γ-CD/E3 and β-CD/E3 were determined in two different environments [89]. The retention factor for α-, β-, and γ-CD/E3 complexes and their HP-CD derivatives were measured using HPLC at 0 and 60 °C in MeCN/water (35% *v*/*v*). The differences of the retention factors point out the importance of the temperature in the HPLC studies. γ-CD/E3 and its HP-derivative have been shown to be the most effective CDs for the host-guest complexation [103].

SBE-β-CD/E3 was extensively studied in silico using the quantum chemistry semi-empirical PM3 method. The heat of formation, dipole moments, stabilization energies, and the capillary electrophoresis migration time for the PM3-optimized geometry of the E3 inclusion complex were calculated. A correlation between migration time and the heat of formation was observed, similar to SBE-β-CD/E2 complexes. It suggests that more stable complexes have a longer migration time in capillary electrophoresis. The theoretical in silico calculations suggest that a CD/E3 complex may exist with either the A or D ring in the cavity while migrating [107]. The presence of β-CD facilitates the separation of E3 from other estrogens using HPLC at specific temperatures. This is caused by shorter retention times after the addition of the CD [104]. When E1, E2 (both stereoisomers), E3, and E4 have to be separated using HPLC, the addition of β-CD and a temperature at exactly 26 or 47 °C enables excellent separation [105]. Similar to E2, an enhancing effect on estrogen separations in MEKC using β-CD and γ-CD as additives was shown for E3. Due to the addition of CD, E3 was more easily separated from another nine different estrogens. A separation using α-CD was also performed and the results showed no successful separation. This could be caused by the cavity of α-CD being not large enough to efficiently guest estrogens [106].

#### 4.6.2. Increased Solubility Resulting from Complexation

Another series of interesting studies presented the results of measuring the solubility of E3, either as a free molecule or complexed with β-CD-branched derivatives using HPLC. Initially, Gal-β-CD, Man-β-CD, and Glc-β-CD were found to increase the solubility of E3 significantly more than native β-CD. That is why in vivo experiments on rats were performed for Gal-β-CD/E3, Man-β-CD/E3, and Glc-β-CD/E3. First, the authors focused on the metabolism and safety of non-complexed Gal-β-CD, Man-β-CD, and Glc-β-CD. After that, they formed complexes and introduced them in large oral doses into rats. As a result, the pharmacokinetic parameters of the complexed E3 were established. The complexation increased the water solubility of the water-insoluble E3. This resulted in an increase in gastrointestinal absorption. The absorption efficiency values were very similar for each studied β-CD derivative [120]. These rather unusual E3/β-CD derivatives complexes not only showed a much higher solubility, but also seemed to be significantly less toxic for living organisms than the other complexes [121]. In the next study, the Glc-α-CD/E3, maltosyl-α-CD/E3, maltotriosyl-α-CD/E3, Glc-β-CD/E3, maltosyl-β-CD/E3, maltotriosyl-β-CD/E3, Glc-γ-CD/E3, maltosyl-γ-CD/E3, and maltotriosyl-γ-CD/E3 complexes were studied. In these cases, the increased solubility for the E3, caused by complexation, was shown [122]. After successfully improving the solubility of E3 by the application of branched derivatives of β-CD, the authors decided to take advantage of α- and γ-CD derivatives. They found that all of the applied γ-CD derivatives formed stable complexes with E3, which resulted in the highly improved solubility of this estrogen. However, none of the used α-CD derivatives were found to form complexes with E3 [122]. A similar study was performed for 2Glc-β-CD/E3. In this CD derivative, the OH groups of β-CD were substituted by disaccharides units. The results were compared with native α-CD/E3, β-CD/E3, and their monoglucosyl derivatives. The authors did not succeed in preparing the stable complexes with α-CD. As anticipated, the complexation with β-CD enhanced the solubility of E3. As shown earlier, the solubility of estrogen in the complex with glucosyl derivatives was even higher than in that of native β-CD complexes. Similar results to the equilibrium solubility were also observed for stability constants. Higher solubilities of these complexes suggested the injection administration route for the complexed API. To examine their safety, the hemolytic effects of pure CDs and their E3 complexes were studied. On the one hand, the results showed that these CDs at higher concentrations cause an hemolytic effect. On the other hand, stable complexes with E3 demonstrated a very low hemolytic effect [123].

#### 4.6.3. Application of the Complexes

CD/E3 complexes are also the objects of studies focused on the water pollution problem. A study showed that β-CD- and γ-CD-based polymers can form highly effective estrogen filters. The polymers including γ-CD molecules in their structure showed better results in the removal of E3 from water than those containing β-CD. This may be caused by the larger inclusion capacity [87]. The usage of CD in micropollutant removal seemed to be very promising, so larger scale experiments were performed. β-CD crosslinked with epichlorohydrin was used for the preparation of the filter. The results showed a correlation between The binding constant of the complex and the sorption efficacy within the filter. The concentration of E3 (5 μg/L) was lowered by 90% after 5 min. Longer time observations (2 h and 48 h) showed no higher concentration reduction. The same paper showed that CD filters were highly selective in removing selected micropollutants. The authors concluded that this observation makes CD a suitable technology for wastewater management [95]. β-CD crosslinked with epichlorohydrin and mixed with quartz sand, active carbon, or commercial sorbent was also studied at a smaller, lab scale. The results showed the high utility of CDs in water waste treatment [96].

### 4.7. Estetrol (E4)

A highly interesting in vivo study of E4/CD complexes has been recently reported. In this work, E4 (Figure 3) was complexed with HP-β-CD and encapsuled in liposomes. The solubility of E4 linearly increased with the CD concentration until 50 mM, which indicated the formation of a 1:1 molar ratio complex. Such a formulation increased the capability of passing the brain-blood barrier by E4. This has a major effect for dealing with the brain problems of some premature babies mentioned in the paper. The complex showed no cellular toxicity and had good physicochemical properties—suitable for future pharmacological usage. Simultaneously, this work provides evidence of the existence of E4/β-CD complexes [124].

E4 was also used in an HPLC study, where the influence of temperature and the addition of β-CD on the efficiency of the separation was analyzed. The temperature had an major effect when estrogens (six different estrogens in this study) in the presence of β-CD were separated. Temperatures of 47 and 26 °C showed the best selectivity of the process and enabled excellent separation [105].

### 4.8. Equilin and Equilenin

Equilin and equilenin (Figure 3) are horse estrogens, which are not present in human organisms under physiological conditions. Despite that, they have found application as APIs; for example, as one of the ingredients of conjugated estrogens (CEEs) mixtures, which are used in replacement hormone therapies for menopausal symptoms [125]. The influence of β-CD on the chromatographic estrogen separation of mixtures containing equilin and equilenin has been studied. It has been shown that, at subambient temperatures, the selectivity of the chromatographic system was greatly enhanced by the addition of CD [104]. A similar study was performed for equilin, this time using β-CD derivatives such as DM-β-CD and HE-β-CD. The results were similar to the previous work. The influence of CD concentration has also been extensively studied. It was concluded that β-CD, both in a native form but especially its derivatives, is very useful in equilin and E3 chromatographic separations [117]. In a study with five other estrogens, it was shown that the best temperature for equilin separation in the presence of β-CD was either 47 or 26 °C [105].

### 4.9. Tibolone (T)

Tibolone (Figure 3) is a steroidal estrogen. There is very little information reported regarding tibolone-CD complexes. We only found one study describing the formulation of β-CD/T complexes, subsequently studied by means of FTIR and UV spectroscopies. In this study, tibolone was mixed with β-CD in different ratios and some practical leads on the preparation of T/β-CD complexes have been described [126].

## 5. Estrogens That Most Likely Do Not form Host-Guest Complexes with Cyclodextrins

### 5.1. Diensterol

Diensterol (Figure 4) is a nonsteroidal estrogen. To the best of our knowledge, there is only one study demonstrating an attempt to determine the possible impact of β-CD on diensterol. In this study, the authors noted no influence on the extraction properties of this estrogen using cyclodextrins. It was suggested that no complex formation between this molecules occurs [127].

### 5.2. Diethylstilbestrol

Diethylstilbestrol (Figure 4) is a nonsteroidal estrogen. The structure of this molecule is similar to dienesterol. This is the main reason why this estrogen also does not form complexes with cyclodextrins. Similar to diensterol, there is only one study showing an attempt to determine the possible impact of β-CD on this estrogen. In this study, the authors noted no influence on the extraction properties of diethylstilbestrol using cyclodextrins. It was suggested that no complex formation between this molecules occurs [127]. Since the discovery of the toxic effects of diethylstilbestrol, it has largely been discontinued and is now mostly no longer marketed [128]. Therefore, it is unlikely that the complexation of this estrogen with CDs will be extensively studied in the future.

## 6. Estrogens with No Evidence of Forming Host-Guest Complexes with Cyclodextrins

Some estrogens (Figure 4), to the best of our knowledge, have no evidence of existing in an host-guest complex with cyclodextrins. The reasons may be different for each estrogen. Some of them may not form them, while others may have not been studied in this way so far. We can divide these estrogens in two major groups. The non-steroidal estrogens that have no evidence of forming such complexes are: chlorotrianisene, fosfestrol, hexestrol, zeranol, and methallenestril. The steroidal estrogens are: methylestradiol and moxestrol.

**Table 1 ijms-24-08780-t001:** Complexes of estrogens and cyclodextrins and the methods used to analyze them. The abbreviations used above for cyclodextrin names stand for: A—amino (many derivatives), DETA—diethylenetriamine, DM—heptakis(2,6-di-o-methyl), EDA—ethylenediamine, Gal—galactosyl, Glc—glucosyl, HE—hydroxyethyl, HP—hydroxypropyl, 2HP—2-hydroxypropyl, HSES—heptakis-6-sulfoethylsulfanyl-6-deoxy, HTG—heptakis-6-thioglyceryl-6-deoxy-, HTMT—heptakis-6-methylsulfanyl-6- deoxy-2-(2-(2-(2-methoxyethoxy)ethoxy)ethyl)], M—methyl, Man—mannosyl, RM—randomly methylated, S—sulfate, SBE—sulfobutyl ether, TEPA—tetraethylenepentamine, TETA—triethylenetetramine, TM—heptakis(2,3,6-tri-O-methyl), TA—triacetyl, TE—triethyl. The abbreviations used above for analytical methods stand for: E—electrophoresis (also capillary), F—fluorescence spectroscopy, IS—in silico, IV—in vivo, HPLC—high-performance liquid chromatography, MEKC—micellar electrokinetic chromatography, NMR—nuclear magnetic resonance, DSC—differential scanning calorimetry, UV—ultra-violet spectroscopy, CM—confocal microscopy, FM—fluorescence microscopy, FTIR—Fourier-transform infrared spectroscopy, SEM—scanning electron microscopy, TEM—transmitting electron microscopy, HSM—hot stage microscopy, TGA—thermogravimetric analysis, XRD—X-ray diffraction, ND—no data. The host-guest ratio column informs on the ratio values mentioned in the following papers. The value should not be interpreted as the ratio used in the method of analysis.

Estrogen (Guest)	Cyclodextrin (Host)	Method of Analysis	Host:Guest Ratio of the Complex	Source
**ethinyloestradiol**	α-CD	F	1:1	[90]
	β-CD	E	1:1	[82]
		IS	1:1; 1:1	[85,92]
		F	1:1; 1:1; 1:1; 1:1	[81,85,90,91]
		NMR	1:1; 1:1; 1:1	[85,87,119]
		HPLC	ND; 1:1	[84,88]
		FTIR	1:1	[85]
		UV	1:1	[85]
	EDA-β-CD	UV	1:1	[83]
		HPLC	1:1	[83]
	DM-β-CD	F	1:1	[90]
	HP-β-CD	HPLC	1:1; 1:1	[83,85]
		F	1:1; 1:1	[85,90]
		SEM	1:1	[83]
		UV	1:1; 1:1	[83,85]
		IS	1:1	[85]
		FTIR	1:1	[85]
	2HP-β-CD	E	1:1	[82]
	DETA-β-CD	UV	1:1	[83]
		HPLC	1:1	[83]
		NMR	1:1	[83]
	TETA-β-CD	UV	1:1	[83]
		HPLC	1:1	[83]
	TEPA-β-CD	UV	1:1	[83]
		HPLC	1:1	[83]
	γ-CD	E	1:1	[82]
		IS	1:1	[92]
		F	1:1; 1:1	[90,91]
		HPLC	1:1	[88]
		NMR	1:2	[87]
	2HP-γ-CD	E	1:1	[82]
	S-β-CD	F	1:1	[90]
	HE-β-CD	F	1:1	[90]
	M-β-CD	F	1:1	[90]
**estradiol**	HP-β-CD in liposomes	TEM	1:1	[97]
	HP-β-CD	UV	1:1; 1:1	[101,108]
		F	1:1; 1:1	[90,101]
		IV	ND	[111]
		HPLC	ND	[103]
		IS	1:1	[101]
		NMR	1:3	[109]
		DSC	1:3	[109]
	SBE-β-CD	E	ND	[107]
		IS	ND	[107]
	β-CD	UV	1:1	[108]
		IS	1:1	[92]
		IV	1:1; 1:1; 1:2;	[97,98,112,113]
		FTIR	ND	[99]
		F	1:1; 1:1; 1:1	[90,91,101]
		FM	ND	[99]
		XRD	ND; 2:1	[99,100]
		HPLC	ND; ND; ND	[103,104,105]
		MEKC		[106]
		UV	1:1; 1:1	[101,108]
		DSC	ND; 2:1; 1:2; 1:1	[99,100,112,113]
		NMR	1:1; 2:1; ND	[87,100,102]
		HSM	2:1	[100]
		TGA	2:1	[100]
	RM-β-CD	IS	1:1	[101]
		NMR	1:1, 1:2,4	[109]
		DSC	1:1, 1:2,4	[109]
	HSES-β-CD	HPLC	1:1	[108]
	HTG-β-CD	HPLC	1:1	[108]
	HTMT-β-CD	HPLC	1:1	[108]
	A-β-CD (many options)	CM	ND	[110]
	γ-CD	HPLC	ND	[110]
		MEKC	ND	[106]
		NMR	1:2; 1:1; ND	[87,100,102]
		F	1:1; 1:1	[90,91]
		IS	1:1	[92]
	α-CD	HPLC	ND	[103]
		MEKC	ND	[106]
		F	1:1; ND	[90,102]
		NMR	ND	[102]
	HP-γ-CD	HPLC	ND	[103]
	HP-α-CD	F	ND	[102]
		HPLC	ND	[103]
	S-β-CD	F	1:1	[90]
	HE-β-CD	F	1:1	[90]
	M-β-CD	F	1:1	[90]
	DM-β-CD	F	1:1	[90]
**estriol**	α-CD	F	1:1	[90]
		HPLC	ND; 1:1; 1:1	[103,122,123]
		MEKC	ND	[106]
	HP-α-CD	HPLC	ND	[103]
	β-CD	NMR	1:1	[119]
		HPLC	ND; 1:1; ND; ND; ND; 1:1; 1:1	[84,88,103,104,105,122,123]
		MEKC	ND	[106]
		F		[90,91]
		IS	1:1; 1:1	[92]
	γ-CD	F	1:1	[91]
		IS	1:1	[92]
		HPLC	1:1; ND; 1:1	[88,103,122]
		MEKC	ND	[106]
	S-β-CD	F	1:1	[90]
	HE-β-CD	F	1:1	[90]
	M-β-CD	F	1:1	[90]
	HP-β-CD	F	1:1	[90]
		HPLC	1:1	[103]
	DM-β-CD	F	ND	[90]
	HP-γ-CD	HPLC	1:1	[103]
	SBE-β-CD	E	ND	[107]
		IS	ND	[107]
	Glc-β-CD	IV	ND	[120]
		HPLC	ND; 1:1; 1:1	[121,122,123]
		DSC	1:1	[123]
	Man-β-CD	IV		[120]
		HPLC	ND	[121]
	Gal-β-CD	IV	1:1	[120]
		HPLC	ND	[121]
	Maltosyl-β-CD	HPLC	1:1	[122]
	Maltotriosyl-β-CD	HPLC	1:1	[122]
	Glc-α-CD	HPLC	1:1; 1:1	[122,123]
	Glc-γ-CD	HPLC	1:1	[122]
	Maltosyl-α-CD	HPLC	1:1	[122]
	Maltosyl-γ-CD	HPLC	1:1	[122]
	Maltotriosyl-α-CD	HPLC	1:1	[122]
	Maltotriosyl-γ-CD	HPLC	1:1	[122]
	2Glc-β-CD	HPLC	1:1	[123]
**chlorotrianisene**	ND	ND		ND
**estrone**	α-CD	F	1:1	[90]
		HPLC	ND	[103]
		MEKC	ND	[106]
	β-CD	HPLC	ND; 1:1; ND; ND; ND; ND	[84,88,103,104,105,117]
		IS	1:1	[92]
		F	1:1	[90]
		MEKC	ND	[106]
	γ-CD	F	1:1	[90]
		IS	1:1	[92]
		HPLC	1:1; ND	[88,103]
		MEKC	ND	[106]
	S-β-CD	F	1:1	[90]
	HE-β-CD	F	1:1	[90]
		HPLC	ND	[117]
	DM-β-CD	F	1:1	[90]
		HPLC	ND	[117]
	M-β-CD	F	1:1	[90]
	HP-β-CD	F	1:1	[90]
		HPLC	ND	[103]
	HP-α-CD	HPLC	ND	[103]
	HP-γ-CD	HPLC	ND	[103]
	SBE-β-CD	E	ND	[107]
		IS	ND	[107]
	E1-β-CD	IV	1:1; 1:2	[118]
		NMR	1:1; 1:2	[118]
		TEM	1:1; 1:2	[118]
		SEM	1:1; 1:2	[118]
**promestriene**	ND	ND		ND
**dienestreol**	ND	ND		ND
**diethylstilbestrol**	ND	ND		ND
**methallenestril**	ND	ND		ND
**moxestrol**	ND	ND		ND
**tibolone**	β-CD	FTIR	ND	[126]
		UV	ND	[126]
**2-hydroxyestrone**	SBE-β-CD	IS	ND	[107]
		E	ND	[107]
	β-CD	HPLC	ND	[117]
	DM-β-CD	HPLC	ND	[117]
	HE-β-CD	HPLC	ND	[117]
**4-hydroxyestrone**	β-CD	HPLC	ND	[117]
		MEKC	ND	[106]
	DM-β-CD	HPLC	ND	[117]
	HE-β-CD	HPLC	ND	[117]
	α-CD	MEKC	ND	[106]
	γ-CD	MEKC	ND	[106]
**16α-hydroxyestrone**	β-CD	HPLC	ND	[117]
		MEKC	ND	[106]
	DM-β-CD	HPLC	ND	[117]
	HE-β-CD	HPLC	ND	[117]
	α-CD	MEKC	ND	[106]
	γ-CD	MEKC	ND	[106]
**2-metoxyestrone**	α-CD	MEKC	ND	[106]
	β-CD	MEKC	ND	[106]
	γ-CD	MEKC	ND	[106]
**2-methoxyestradiol**	SBE-β-CD	IS	ND	[107]
		E	ND	[107]
	DM-β-CD	UV	1:1; 2:1	[116]
		HSM	1:1; 2:1	[116]
		TGM	1:1; 2:1	[116]
		DSC	1:1; 2:1	[116]
		FTIR	1:1; 2:1	[116]
		XRD	1:1; 2:1	[116]
		IS	1:1; 2:1	[116]
	TM-β-CD	UV	1:1; 2:1	[116]
		HSM	1:1; 2:1	[116]
		TGM	1:1; 2:1	[116]
		DSC	1:1; 2:1	[116]
		FTIR	1:1; 2:1	[116]
		XRD	1:1; 2:1	[116]
		IS	1:1; 2:1	[116]
	α-CD	UV	1:1; 2:1	[116]
		MEKC	ND	[106]
	β-CD	UV	1:1; 2:1	[116]
		MEKC	ND	[106]
	γ-CD	UV	1:1; 2:1	[116]
		MEKC	ND	[106]
	RM-β-CD	UV	1:1; 2:1	[116]
	TM-α-CD	UV	1:1; 2:1	[116]
	HP-β-CD	UV	1:1; 2:1	[116]
	TA-β-CD	UV	1:1; 2:1	[116]
	TA- γ-CD	UV	1:1; 2:1	[116]
	TE- γ-CD	UV	1:1; 2:1	[116]
**16-Keto-17β-estradiol**	α-CD	MEKC	ND	[106]
	β-CD	MEKC	ND	[106]
	γ-CD	MEKC	ND	[106]
**2-hydroxyestradiol**	α-CD	MEKC	ND	[106]
	β-CD	MEKC	ND	[106]
		IV	ND; 1:1	[112,113]
	γ-CD	MEKC	ND	[106]
**4-hydroxyestradiol**	α-CD	MEKC	ND	[106]
	β-CD	MEKC	ND	[106]
		IV	ND; 1:1	[112,113]
	γ-CD	MEKC	ND	[106]
**equilin**	β-CD	HPLC	ND; ND	[104,105]
	DM-β-CD	HPLC	ND	[117]
	HE-β-CD	HPLC	ND	[117]
**equilenin**	β-CD	HPLC	ND	[104]
**estetrol**	β-CD	HPLC	ND	[105]
	HP-β-CD	IV	1:1	[124]
**methylestradiol**	ND	ND		ND
**fosfestrol**	ND	ND		ND
**hexestrol**	ND	ND		ND
**zeranol**	ND	ND		ND

## 7. Conclusions

Because estrogens have a low polarity, they can interact with some cyclodextrins’ hydrophobic cavities to create inclusion complexes if their geometric properties are compatible. As shown in this review, estrogen-CD complexes have been widely applied in several chemical fields for various objectives. CDs have been used as estrogen solubilizers and absorption boosters in pharmaceutical formulations, as well as in chromatographic and electrophoretic procedures for their separation and quantification. Other applications include the removal of endocrine disruptors from environmental materials, the preparation of samples for mass spectrometric analysis, and solid-phase extractions based on complex formation with CDs. Through the creation of complexes, CDs were also employed to block the negative effects of estrogens in ambient water.

Although a lot of inclusion complexes between estrogens and cyclodextrins have already been obtained and deeply characterized, the variety of CDs indicates there is still a lot to discover on this topic. This is especially due to the fact that, as shown in this review, the type of cyclodextrin used for complexation can have an enormous influence on the properties of the formed complex, such as the dissolution boost of the guest molecule, the host:guest ratio, or complex stability. Therefore, we hope that this review will help to plan such experiments and facilitate the comparison of the newly obtained results with already published ones.

## Figures and Tables

**Figure 1 ijms-24-08780-f001:**
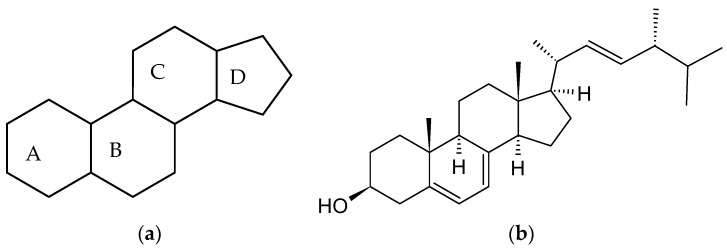
Chemical structure of: (**a**) a gonane unit and (**b**) ergosterol.

**Figure 2 ijms-24-08780-f002:**
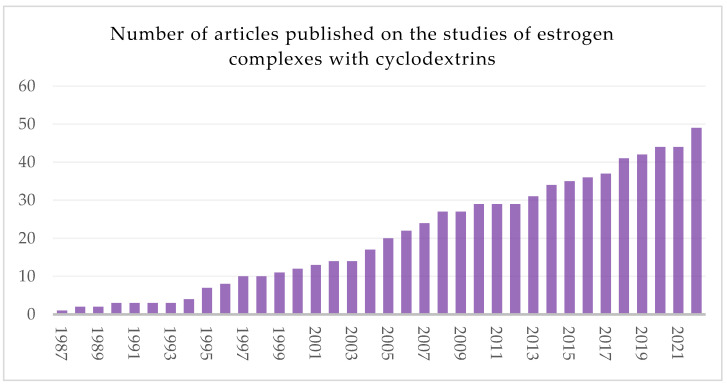
Number of search results for the articles describing the complexes formed between estrogens and cyclodextrins. The graph shows the number of records for query ‘estrogen AND cyclodextrin’ in Scopus and the Web of Science in the period 1987–2022. Each column shows the number of articles in the given year and all years before. For example, the column entitled ‘2010’ depicts the number of articles published in the period 1987–2010, including 2010.

**Figure 3 ijms-24-08780-f003:**
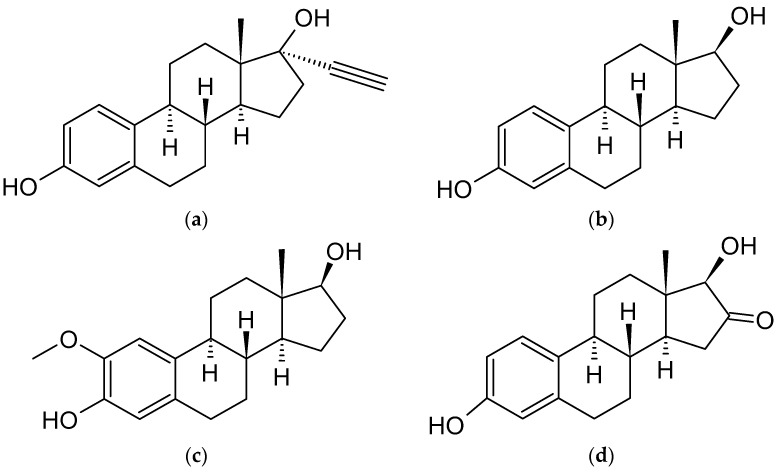
Chemical structures of steroidal estrogen molecules, which form host-guest complexes with CDs: (**a**) ethinylestradiol (EE2), (**b**) estradiol (E2), (**c**) 2-methoxyestradiol (2ME2), (**d**) 16-keto-17β-estradiol (16KE2), (**e**) 2-hydroxyestradiol (2HE2), (**f**) 4-hydroxyestradiol (4HE2), (**g**) estrone (E1), (**h**) 2-hydroxyestrone (2HE1), (**i**) 4-hydroxyestrone (4HE1), (**j**) 16α-hydroxyestrone (16HE1), (**k**) 2-metoxyestrone (2ME1), (**l**) estriol (E3), (**m**) estetrol (E4), (**n**) equilin, (**o**) eqilenin, (**p**) tibolone (T), and chemical structures of steroidal estrogen molecules, which have no evidence of forming host-guest complexes with CDs: (**q**) methylestradiol, (**r**) promestriene, (**s**) moxestrol.

**Figure 4 ijms-24-08780-f004:**
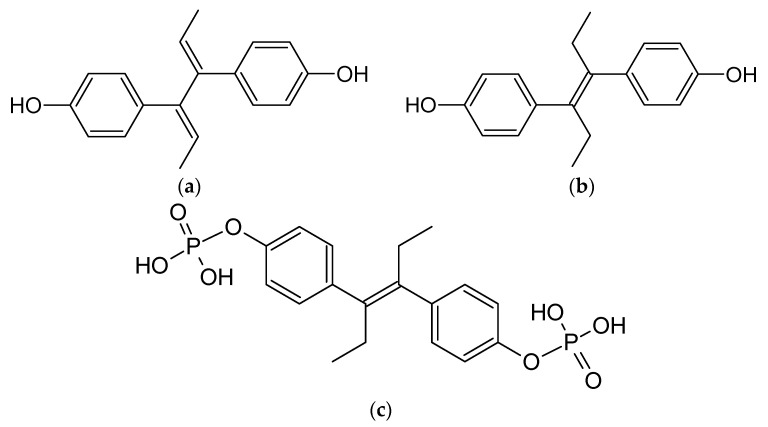
Chemical structures of non-steroidal estrogen molecules, which do not form host-guest complexes with CDs: (**a**) diensterol, (**b**) diethylstilbestrol, and chemical structures of estrogen molecules, which have no evidence of forming host-guest complexes with CDs: (**c**) fosfestrol, (**d**) hexestrol, (**e**) zeranol, (**f**) chlorotrianisene, (**g**) methallenestril.

## Data Availability

Not applicable.

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
