# Peer review of "A Review on Cyclodextrins/Estrogens Inclusion Complexes"

_ijms, 2023, doi:10.3390/ijms24108780_

Round 1

Reviewer 1 Report

The manuscript "A Review on Cyclodextrins/Estrogens Inclusion Complexes" is well-written and it is quite interesting. I like a lot authors explaining the structure and organization of the manuscript in the introduction since it eases the reading. Anyway I have some suggestions and/or request of clarifications

1) The first part of the introduction is written using the italic character format. I think it should be an oversight.

2) line 111: please explain the meaning of the acronym API. It is explained later (line 259) but it should be at the first time the acronym is used.

3) line 213: please specify the meaning of DETA-β-CD and also of other short names of chemical substances in the text. For example look at line 238 DM-β-CD  In literature, there are many different ways of naming the same substance and it is misunderstanding.

4) Many times in the text it is stated that methanol is used to prepare host/guest complexes. Isn't it a problem for human somministration?

5) Line 303: What do you mean 2:1 in form of polymer-based material ? 

6) Line 399: What derivatives are cytotoxic?

7) Line 419: Explain the meaning of HPMC 

8) Line 504: It is written "various CDs". I think it is important to specify in what structural feature ( cavity size, presence of moieties etc) the various CDs differ.

9) Line 567: I suppose that filters are based on CD polymers. What cross linker has been used to link CDs?

10)  General suggestion: At the end of each paragraph there are some lines about the water pollution problem. The authors often refer to reference 98 and repeat the same information. Maybe it is better to write a separate paragraph about methods to remove estrogens from water. In this way, repetitions will be avoided.

11) It is well-known that in the interaction host/guest is not involved only in the CD cavity. This is particularly important in the case of CD polymers where also the networks made of cross-linker and the external of CD play an important role. I think that some information about this kind of interaction should be  added.

Reviewer 2 Report

The submitted manuscript, entitled A Review on Cyclodextrins/Estrogens Inclusion Complexes is extensively summarizing estrogen/CD complexation, their applications in various fields and is introducing the most relevant analytical techniques.

The fact that meanwhile many estrogens are known is reflected in an increasing number of publications in the this field. In contrast with other papers, the authors did not exclusively focus on the most prominent steroidal estrogens but also on various nonsteroidal estrogens and their capability to form CD-complexes, which significantly improve the quality of this review.

The authors clearly pointed out the intention and scope of their review, as mentioned in the introduction: This work is divided into three main sections. In the first one the chemistry and role of estrogens and cyclodextrins is briefly summarized. It was not our intention to make this part extensively long, as there are many recent general reviews focusing on either the estrogens or cyclodextrins. Second part is the main one and is divided into paragraphs describing various complexes of particular estrogens. Finally, the last part concludes the previous ones but also includes some future perspectives in this topic. Furthermore, the authors clearly indicate the distinction between the submitted manuscript and already published reviews.

The consistent structure of the paragraphs facilitates the comparison of the different published results, which makes it more comfortable for the interested readers. In addition, the fluent text is reader friendly but without loss of information.

According the aim of this review, the style, the structure and the completeness of this manuscript I will warmly recommend the publication of this review. 

For final publication English typing errors and mistakes need to be corrected, which should be assisted by a native speaker or professional language translator.  For subject specific errors such as ethynyl/ethinyl or equilienin/equilenin and  in vivo-written sometimes italic and sometimes non-italic, please carefully revise the manuscript.

Reviewer 3 Report

This review (Ms ID: ijms-2398225) first describes the biological functions and applications of estrogens as therapeutic agents, then introduces cyclodextrins as well as their inclusion complexes with steroids, subsequently focuses on cyclodextrin inclusion complexes with different types of estrogen. This paper is well organized and I found no similar reviews in the Web of Science database. My comments are as follows:

Major points:

1.     It will be better to merge Figures 1-14 into 1-2 figures and to briefly introduce them in the first part of the second section (e.g, add a new part ‘2.1. Chemical structures of different estrogens’ and change the original 2.1. part into 2.2. part). One of the advantages of this arrangement is to make it easy for readers to compare the differences among different estrogens.

2.     Some parts (e.g., parts 4.1., 4.2., 4.5., and 4.6.) of the main text are pretty lengthy and complicated. It will be better to use some subheadings to let readers readily understand the logical relationship between paragraphs in those parts.

3.     The information in Table 1 is too limited. More information, such as the optimal host:guest ratio, the mechanisms of host-guest interactions, and other parameters evaluated in most studies, should be included in the table or in a separate table.

4.     Figure 15 should be moved to the Introduction section. Moreover, the information on database and searching method should be provided in the figure legend.

Minor points:

5.     Lines 22-30: All words are in italics. It is unnecessary.

6.     Line 741: ‘4.9Tibolone. (T)’ should be ‘4.9. Tibolone (T)’.

7.     The full name of API appears in line 260 on page 6 whereas the first API appears in line 111 on page 3. All abbreviations should be defined at the first mention.

The English language is acceptable.

Round 2

Reviewer 3 Report

All my concerns have been address. I just have a suggestion. The authors should make the picture size of of each molecule in Figs. 3 & 4 smaller. So, all molecules in each figure, as well as figure caption, can be showed together on one page.